# Chasing high and stable wheat grain mineral content: Mining diverse spring genotypes under induced drought stress

**Yuzhou Lan, Ramune Kuktaite, Aakash Chawade, Eva Johansson** [ID]*

Department of Plant Breeding, The Swedish University of Agricultural Sciences, Lomma, Sweden

* eva.johansson@slu.se

**Data Availability Statement:** All data is available in the Supplementary S2 Table.

**Funding:** This research was funded by Trees and Crops for the Future (TC4F) and SLU Grogrund.

## Abstract

Climate change-induced drought has an effect on the nutritional quality of wheat. Here, the impact of drought at different plant stages on mineral content in mature wheat was evaluated in 30 spring-wheat lines of diverse backgrounds (modern, old and wheat-rye-introgressions). Genotypes with rye chromosome 3R introgression showed a high accumulation of several important minerals, including Zn and Fe, and these also showed stability across drought conditions. High Se content was found in genotypes with chromosome 1R. Old cultivars (K, Mg, Na, P and S) and 2R introgression lines (Fe, Ca, Mn, Mg and Na) demonstrated high mineral yield at early and late drought, respectively. Based on the low nutritional value often reported for modern wheat and negative climate effects on the stability of mineral content and yield, genes conferring high Zn/Fe, Se, and stable mineral yield under drought at various plant stages should be explicitly explored among 3R, 1R, old and 2R genotypes, respectively.

## Introduction

Global food security is highly challenged both in terms of adequacy and nutritional value as a result of an increasing world population, ongoing climate change and unstable human conditions due to migration, poverty and conflicts (wars). This has led to a sharp increase in the global prevalence of undernourishment from 7.9% in 2019 to 9.3% in 2020 and remained at a high level (9.2%) until 2022 [1]. Intake of mineral nutrients is essential in this context as they contribute to the absorption and function of vitamins by the human body [2]. Among the minerals, zinc (Zn) and iron (Fe) have been described as the ones of utmost importance to human health. Zinc deficiency has been recognized as a threat to public health as it links broadly to weakened immunity, liver disease and diarrheal problems [3,4], while the anemia caused by Fe deficiency is the most common health issue worldwide [5,6]. Additionally, selenium (Se) plays a critical role for the human immune function, brain function, male fertility and type-2 diabetes risks [7,8]. Furthermore, calcium (Ca) is widely involved in life processes in cells and bone health [9,10]; copper (Cu) is tightly intertwined with the metabolism of other minerals, i.e. Cu deficiency results in Fe deficiency [11]; magnesium (Mg) has been reported to have a function of activating vitamin D in the human body [12]; manganese (Mn) is essential for the activation of metalloenzymes [13], and potassium (K) and sodium (Na) actively play a role in neurologic and muscular systems, where the Na-K flux on membranes is known to drive nerve impulses by

The funders had no role in study design, data collection and analysis, decision to publish, or preparation of the manuscript.

**Competing interests:** The authors have declared that no competing interests exist.

changing the electrical potential [14]; phosphorus (P) intake at high levels is associated with an increased risk of cardiovascular disease [15,16]; Dietary sulfur (S) is supplied to humans from various sources e.g. vegetables (allium and brassica species), legume crops and animal-based products. It plays a role in protein synthesis in the form of the two primary S-containing amino acids methionine and cysteine [17,18] and is stored as a key metabolite glutathione [19].

In plants, mineral elements are mainly absorbed as ions from the soil, and their content and composition play a key role in plant growth and reproduction [20]. The amount of Zn is extremely important for the enzyme activity of the plants, as it is present in six important plant enzyme classes i.e., oxidoreductases, transferases, hydrolases, lyases, isomerases and ligases [21]. Iron is important for plants as it accumulates in chloroplasts in the green leaves, based on its indispensable role in photosynthetic activities [22,23]. Differently from Zn and Fe, Se is not an essential element for plants although an appropriate amount of Se contributes to plant growth and stress tolerance [24–26]. In addition, other mineral elements e.g. Ca, Cu, Mg, Mn, K, Na, P and S, are also required by the plant to sustain most physiological processes such as photosynthesis, enzyme activation, protein synthesis and pollen formation [23,27]. Thus, mineral accumulation is closely related to both the healthy growth and nutritional value of a crop.

Drought, the major abiotic yield-limiting factor [28], is also known to impact the process of nutrient uptake of a plant due to the drought-induced physiochemical (nutrient mobility and absorbance) changes in soils [29]. Water deficit conditions in the soil are known to negatively affect the mineral uptake in the plants by i) impaired uptake power in the root due to inhibited activity and ii) limited ion diffusion rate due to the low moisture level. However, drought conditions are also known to reduce the content of mineral elements in the plant as transpiration rate and membrane permeability are restricted [30,31]. By the predicted increase in drought events due to climate change, an increased understanding of the effects of drought on mineral content in plants is urgently needed to achieve a high and stable nutritional content in crops.

Wheat (*Triticum aestivum*), one of the three major cereal crops, is feeding the world population with a share of approximately 20% of the calories and proteins [32]. Because of the high daily consumption, wheat products are a crucial source of nutrition for humans. However, similarly to other crops wheat is facing an increasing number of drought spells because of climate change [33]. The timing of the drought spells contributes with different effects to the wheat, e.g. early drought resulted in the inhibition of morphologic traits while late drought restricted the yield [34]. The mineral concentration of the wheat grain is known to be determined by genetics, the environment and their interactions, although a general decrease in the mineral nutritional value of wheat grain has been reported as a result of breeding selections [35]. Drought during field conditions has been found to contribute to a significant increase in grain Zn concentration [36].

This study aimed to deepen the understanding of the impacts of early and late drought stress on wheat grain mineral composition (11 mineral elements). Combined with previously obtained grain yield data, the amount of each element was calculated to identify the single-plant-based nutritional value of wheat from different genetic backgrounds. Another aim was to identify genetic resources of high and stable nutritional value in terms of mineral amount for breeding programs using a wide array of wheat materials.

## Materials and methods

### Plant materials

A total of 30 spring wheat (*Triticum aestivum L.*) genotypes including modern (n = 5), old (n = 5), introgression wheat with rye chromosome 1R (n = 5), 1RS (n = 5), 2R (n = 5) and 3R (n = 5), selected from a previous investigation [34] were used in this study (S1 Table).

## Growing conditions and drought treatments

Similarly as has been described previously [34], pPlants were grown under controlled climatic conditions from April to September, 2020 in the Biotron at the Swedish University of Agricultural Sciences in Alnarp, Sweden, using natural light and hourly-regulated temperature and humidity derived from the average climate data of Malmö, Sweden during the period of 2010–2019 (Swedish Meteorological and Hydrological Institute, SMHI). This experiment used 2.5 L pots filled with soil (product name: Exklusiv Blom och Plantjord 50 liter; article number: 1640; pH: 5.5–6.5) containing 50% of low humified peat, 33% of highly humified peat, 7% of gravel, 5% of leca balls (2–6 mm), 5% of clay with silicon provided by Emmaljunga Torvmull AB (https://www.emmaljungatorvmull.se/), Sweden. The total set of 30 genotypes was subjected to each of the three growing conditions i.e. one control and two drought treatments (EDS: early drought stress; LDS: late drought stress) with three biological replicates used within each condition. Plants grown under control were watered every second day until spike maturity. Both drought treatments were applied in the form of water-withholding, with EDS starting 30 days after sowing and lasting 4 weeks, and LDS starting 60 days after sowing and lasting 2 weeks.

## Sample preparation

From each growing condition, three biological replicates of each genotype were sampled, resulting in a total of 270 samples. All grain samples were oven-dried at 40°C for 24 h and then milled for 30 s into flour (mixer mill 400 MM, RETSCH). For digestion, 150 mg of each flour sample was mixed with 3 ml of nitric acid (69–70%, J.T.Baker-instra analyzed) and then the samples were subjected to autoclave (GETINGE, Sweden) conditions; 121°C, 200 Kpa for 30 min. Thereafter, the digested samples were cooled down to room temperature, and then 27 ml of Milli-Q water was added to dilute each solution 10 times. Finally, a total of 10 ml of the solution of each sample was collected and used for mineral analysis.

## Mineral determination and mineral yield calculation

Concentrations of Zn, Se and Mn were determined by inductively coupled plasma mass spectrometry (ICP-MS, Aurora Elite, Bruker, U.S.) while concentrations of Ca, Cu, Fe, K, Mg, Na, P and S were determined by inductively coupled plasma optical emission spectrometry (ICP-OES, Optima 8300, Perkin-Elmer, U.S.). To describe the amounts of minerals provided by the grains of a plant (mg/plant), the single-plant-based yield of each mineral was calculated by multiplying mineral concentration and grain yield (grain weight per plant). Standards used in the analysis were atomic spectrometry standards from Perkin-Elmer, SPEX, AccuStandard and Merck. Calibration of the ICP-OES instrument was done by using a mixed multicomponent standard at three concentrations within the factor of 50 and calibration was maintained with independent standards. The detection limit used was three times the standard deviation based on multiple determination of the blanks treated as the sample, were blanks were treated identically and together with the samples. All the mineral concentration, grain yield and mineral yield data can be found in S2 Table.

## Data analysis

All statistical analyses were done in RStudio [37]. A two-way analysis of variance (ANOVA) was performed to detect variations between treatments and among genotypes. The pairwise comparisons (LSD post-hoc test) between genotype groups (modern, old, 1R, 1RS, 2R and 3R) and between treatments (C, EDS and LDS) were performed using the R package 'agricolae'. The linear regression presented in scatter plots were computed and visualized using R package

'ggplot2' and 'ggpmisc'. Principal component analysis (PCA) was computed using R packages 'ggfortify' and 'rgl'. The additive main effects and multiplicative interaction (AMMI) was performed using the R package 'metan' to identify genotypes with high and stable nutritional value.

## Results

### Genotypic variations in minerals in relation to drought stresses

ANOVA showed a highly significant effect of both genotype and drought treatment on the mineral grain concentration and yield of most of the 11 mineral elements (Zn, Fe, Se, Ca, Cu, K, Mn, Mg, Na, P and S) evaluated (S3 Table).

The PCA clearly divided the genotypes based on the three treatments (control, EDS and LDS) along the first principal component (PC1) axis, accounting for 50.1% and 75.0% of the variation for mineral concentration (Fig 1A) and mineral yield (Fig 1B), respectively. However, the mineral concentration largely overlapped under control and EDS, indicating a lack of impact from EDS on grain mineral concentration. For grain mineral concentration, the samples grown under LDS, generally showed more positive PC1 values than samples grown under control and EDS. The concentrations of all minerals, with the exception of Mn, were also located with positive PC1 values, indicating a positive correlation between mineral concentration and LDS treatment (Fig 1A). Also for mineral yield, LDS samples were clearly differentiated along PC1 with more positive values than for control and EDS samples (Fig 1B). However, the mineral yield of the different minerals was in this case clustered with negative PC1 values, indicating a negative correlation between mineral yield and LDS treatment (Fig 1B).

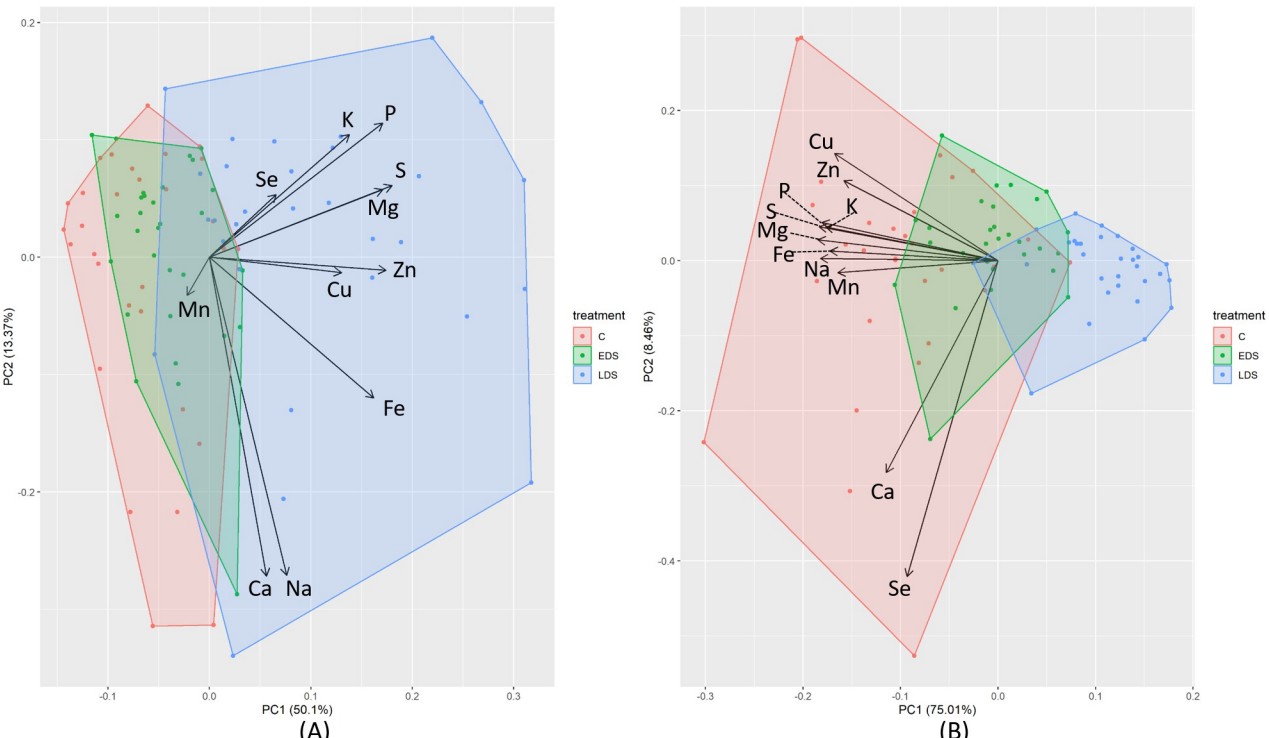

**Fig 1.** Biplots of principal component analysis (PCA) for the (A) grain concentration and (B) mineral yield of Zn, Fe, Se, Ca, Cu, K, Mn, Mg, Na, P and S of genotypes studied under control (C), early drought stress (EDS) and late drought stress (LDS) conditions.

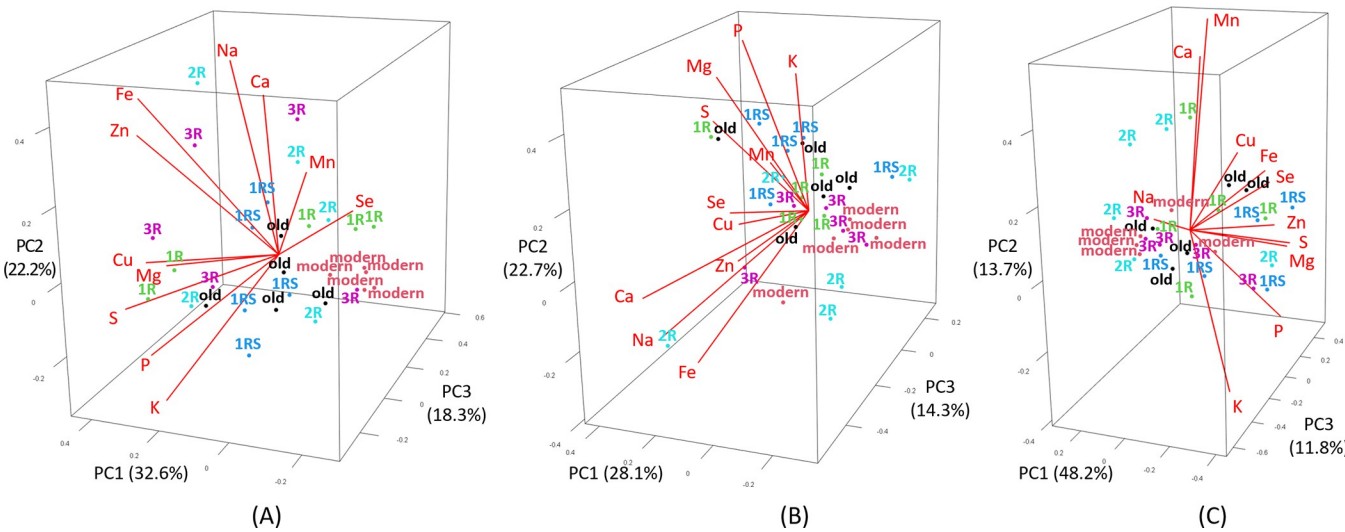

**Fig 2.** Principal component analysis (PCA) for grain concentrations of Zn, Fe, Se, Ca, Cu, K, Mn, Mg, Na, P and S in different genotype groups (modern = approved cultivars and breeding lines received from the breeding company Lantmännen, old = old Swedish cultivars released from 1928 to 1990, 1R, 1RS, 2R and 3R = Introgressions of chromosome 1R, 1RS, 2R and 3R) under (A) control, (B) early drought stress and (C) late drought stress.

### Relationships between mineral concentration/yield and genotype groups

The drought treatments affected the genotypes of various groups differently. Basically, all modern genotypes were found consistently with lower grain mineral concentrations than the other genotype groups across all the three treatments applied (control, EDS and LDS; Fig 2), indicating a relatively poor grain mineral nutrition of modern wheat compared with the other genotype groups. Differently, some old genotypes were found with high grain concentrations of Fe and Se, especially under LDS (Fig 2C). Furthermore, 1R genotypes showed a high grain concentration of Se and Mn under control (Fig 2A) while 1RS genotypes showed a high grain concentration of P, K and Mg under EDS (Fig 2B) and of Zn, Fe, Se, Mg, P and S under LDS (Fig 2C).

For mineral yield, modern genotypes generally showed the lowest values as compared to the other genotype groups (Fig 3A), but some modern genotypes were found with a high yield of some minerals under EDS (Cu, Fe; Fig 3B) and LDS (K; Fig 3C), which might be the result of a high grain yield of modern genotypes. Furthermore, some old genotypes showed high mineral yield for K, Mn, Mg, Na, P and S under EDS (Fig 3B). Also, 1R (Se and Mn) and 3R (Zn, Fe, Cu, K, Mg, P and S) genotypes were found with high mineral yield for different minerals under control (Fig 3A), while 2R genotypes were found with high mineral yield for Fe, Ca, Cu, Mn, Mg, Na, P and S under LDS (Fig 3C).

### Grain concentration and mineral yield of Zn and Fe

Both grain Zn and Fe concentration and mineral yield of these components varied based on drought stress at different development stages (EDS and LDS) but also based on genotype groups (modern, old, 1R, 1RS, 2R and 3R). The Zn grain concentration was generally higher under LDS than under control for most of the genotype groups, with the exception of 2R genotypes (Fig 4A). Similarly, most genotype groups showed a higher Fe concentration under LDS than under control, with the exception of 2R and 3R, while only the modern genotype group displayed an increase in Fe concentration under EDS (Fig 4B).

A significantly higher Zn concentration was found for 3R genotypes (21.25 mg/kg) than most other genotype groups except the 2R genotypes under control while no difference was

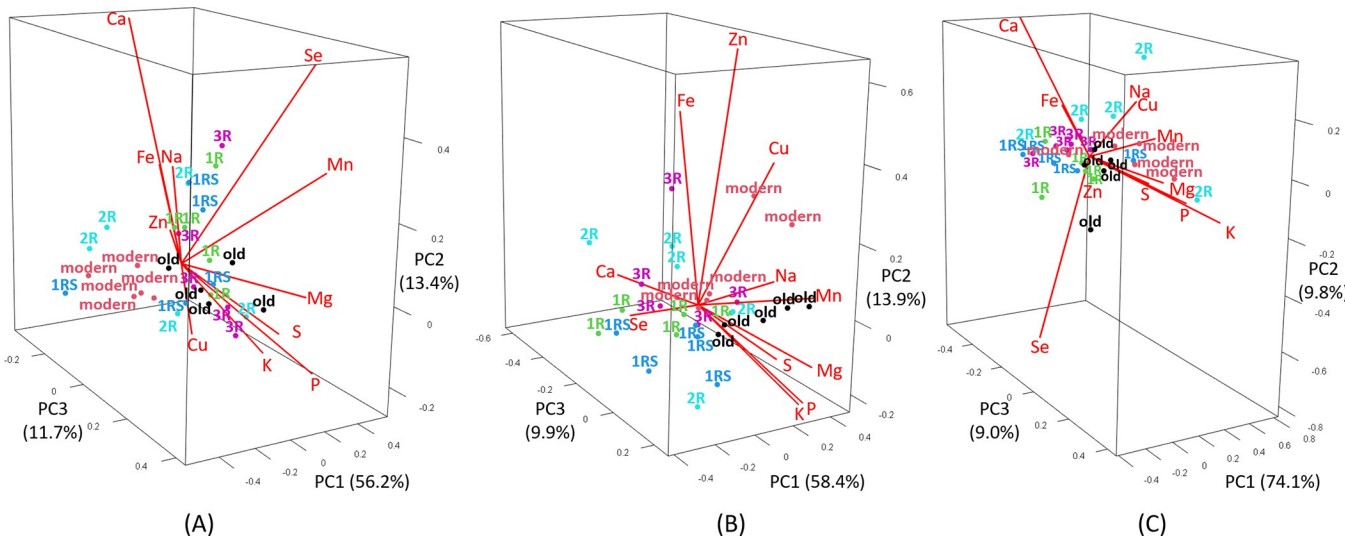

**Fig 3.** Principal component analysis (PCA) for mineral yield of Zn, Fe, Se, Ca, Cu, K, Mn, Mg, Na, P and S in different genotype groups (modern = approved cultivars and breeding lines received from the breeding company Lantmännen, old = old Swedish cultivars released from 1928 to 1990, 1R, 1RS, 2R and 3R = introgressions of chromosome 1R, 1RS, 2R and 3R) under (A) control, (B) early drought stress and (C) late drought stress.

found between genotype groups under EDS and LDS (Fig 4A). Both 2R (21.00 mg/kg) and 3R (21.81 mg/kg) genotypes showed a higher Fe concentration than modern genotypes (12.54 mg/kg) under control. Furthermore, a higher Fe concentration was found in 2R lines as compared to 1RS lines under EDS and in 1RS lines as compared to modern genotypes under LDS (Fig 4B).

No significant change in Zn yield by EDS and LDS treatments as compared to control was obtained for modern and old genotypes (Fig 4C). The Fe yield was significantly increased by EDS in modern genotypes, while LDS resulted in a decrease in Fe yield for both modern and old genotypes (Fig 4D). A decrease in both Zn and Fe yield under EDS as compared to control was obtained for 1R and 3R genotypes, while LDS reduced both Zn and Fe yield for all introgression lines (1R, 1RS, 2R and 3R; Fig 4C and 4D).

Significantly the highest Zn (0.11 mg/plant) and Fe (0.12 mg/plant) yield under control was found for the 3R genotypes (Fig 4C and 4D). Low Zn and Fe yield were obtained from 1R and 1RS lines under both EDS and LDS, while also 3R lines resulted in a low Zn and Fe yield under LDS (Fig 4C and 4D).

## Genotypes with high concentrations and yield of Zn and Fe

A clear positive relationship between Zn and Fe was found for both grain concentration (Fig 5A) and yield (Fig 5D) for the evaluated genotypes under control ($R^2 = 0.63$ and $0.77$, respectively). This positive relationship was decreased ($R^2 = 0.43$ and $0.35$, respectively) under EDS (Fig 5B and 5E), and increased ($R^2 = 0.77$ and $0.81$, respectively) under LDS (Fig 5C and 5F). Corresponding to the PCA results (Figs 2A and 3A), all the modern genotypes were located at the bottom left corner of the plots under control (Fig 5A and 5D), indicating a simultaneously low Zn and Fe in modern genotypes at non-drought conditions.

Two 3R genotypes (250 and 251) and one 2R genotype (258) with both high Zn and Fe grain concentrations (Fig 5A and S1A and S1D Fig), and three 3R genotypes (250, 251 and 256) with high Zn and Fe yield (Fig 5D) were identified under control. The genotypes 258 (2R) and 250 (3R; Fig 5B and S1B and S1E Fig), and the genotypes 250 (3R) and 279 (modern; Fig 5E) displayed high Zn and Fe grain concentration and mineral yield, respectively, under EDS

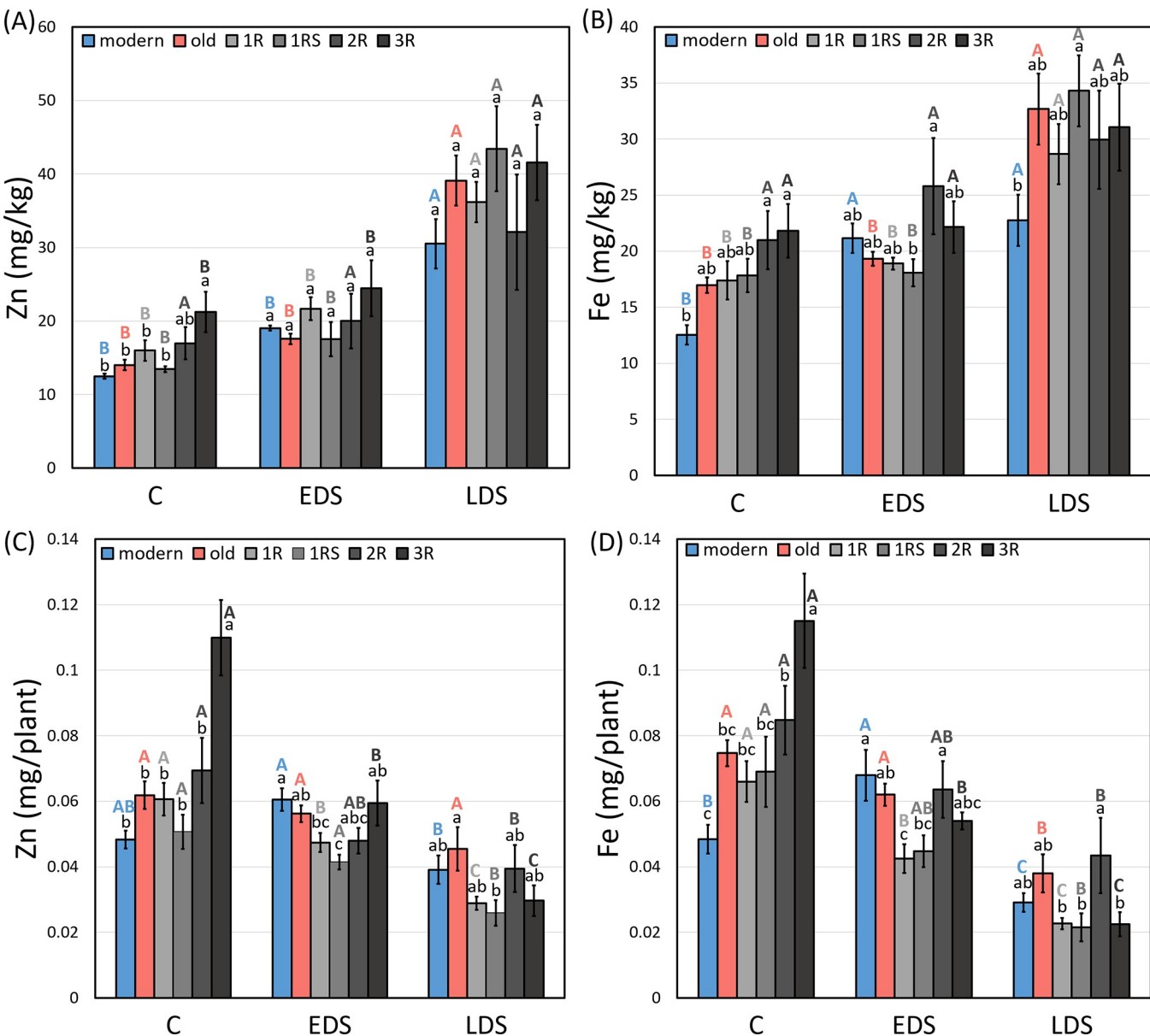

**Fig 4.** The mean grain concentration of (A) Zn and (B) Fe, and mineral yield of (C) Zn and (D) Fe of each genotype group under control (abbreviated as C), early drought (EDS) and late drought stress (LDS). Modern = approved cultivars and breeding lines received from company Lantmännen, old = old Swedish cultivars released from 1928 to 1990, 1R, 1RS, 2R and 3R = Introgressions of chromosome 1R, 1RS, 2R and 3R. Means of the same genotype group between treatments marked by the same capital letters do not differ significantly. Means between different genotype groups within each treatment marked by the same lowercase letters do not differ significantly (LSD post-hoc test at p < 0.05).

conditions. Furthermore, the genotypes 235 (1RS), 245 (2R) and 250 (3R; Fig 5C and S1C and S1F Fig) and the genotypes 207 (old) and 270 (2R; Fig 5F) showed high Zn and Fe grain concentration and mineral yield, respectively, under LDS.

## Stability of the concentration and yield of Zn and Fe

The additive main effects and multiplicative interaction (AMMI) suggested a similar genotype × environment (treatments) interaction pattern between Zn and Fe grain

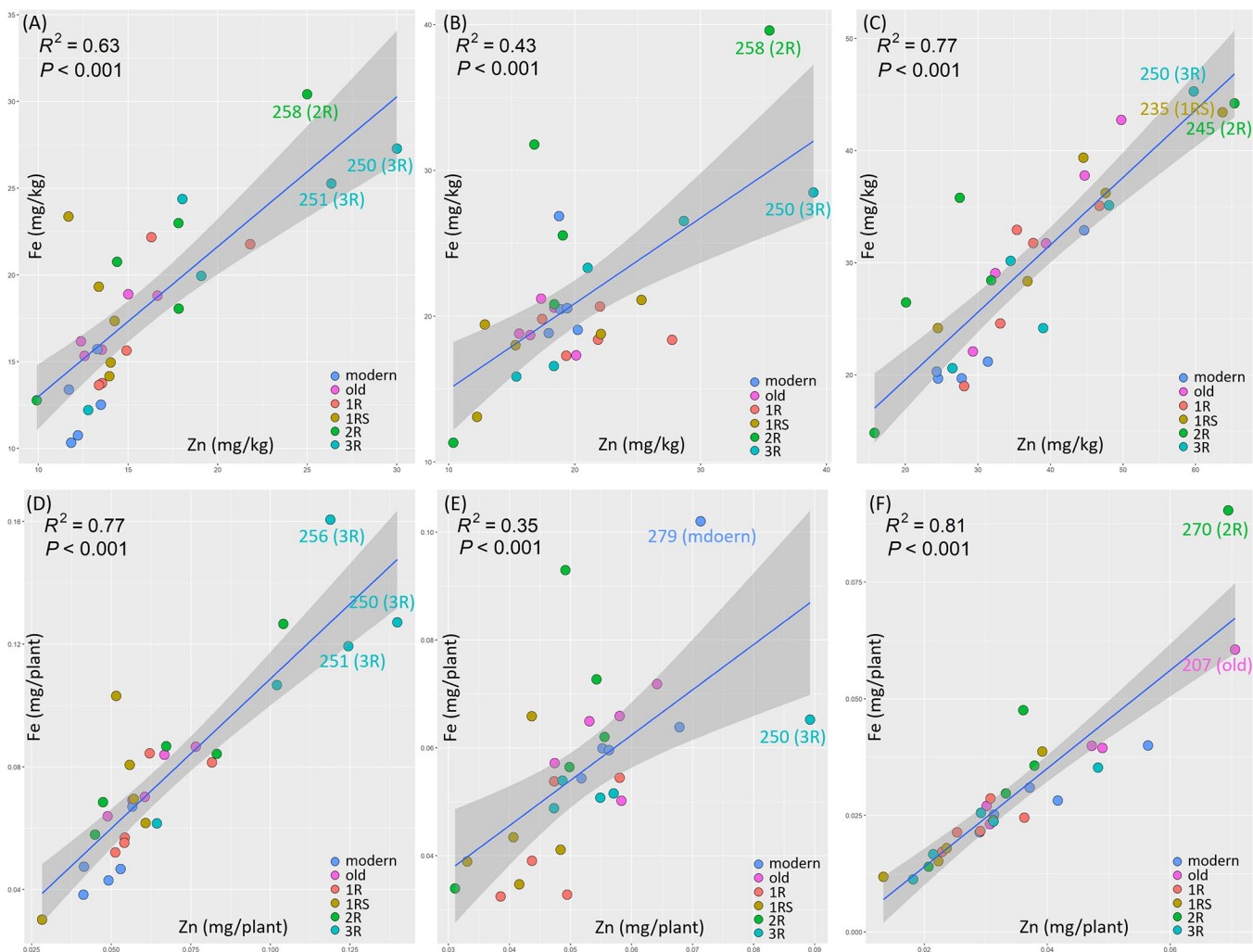

**Fig 5.** Linear regression ($R^2$ = the coefficient of determination) of Zn and Fe grain concentration and mineral yield in genotypes under (A and D) control (abbreviated as C), (B and E) early drought stress (EDS) and (C and F) late drought stress (LDS). Modern = approved cultivars and breeding lines received from company Lantmännen, old = old Swedish cultivars released from 1928 to 1990, 1R, 1RS, 2R and 3R = Introgressions of chromosome 1R, 1RS, 2R and 3R.

concentration (Fig 6A and 6B), as well as between Zn and Fe yield (Fig 6C and 6D). For Zn and Fe concentrations, LDS showed the strongest interaction effect resulting in above-average concentrations while both control and EDS showed the interaction force resulting in below-average concentrations. Among the above-average genotypes, 250 (3R) and 251 (3R) were identified as high-and-stable genotypes for both Zn and Fe concentrations (Fig 6A and 6B). For Zn and Fe yield, the strongest interaction force was identified under control which resulted in above-average values, while LDS resulted in below-average values (Fig 6). Genotypes 250 (3R) and 251 (3R) showed high and stable Zn yield (Fig 6C) while genotypes 250 (3R), 251 (3R), 256 (3R), 258 (2R), 270 (2R) and 271 (2R) showed high and stable Fe yield (Fig 6D).

## Concentration and yield of Se

Basically, no effect of drought stress was found on Se concentration (Fig 7A) while Se yield was decreased in 2R lines by EDS and in all introgression lines (1R, 1RS, 2R and 3R) by LDS as compared to control (Fig 7B). High grain concentration of Se was found in 1R genotypes at control, EDS and LDS conditions (Fig 7A).

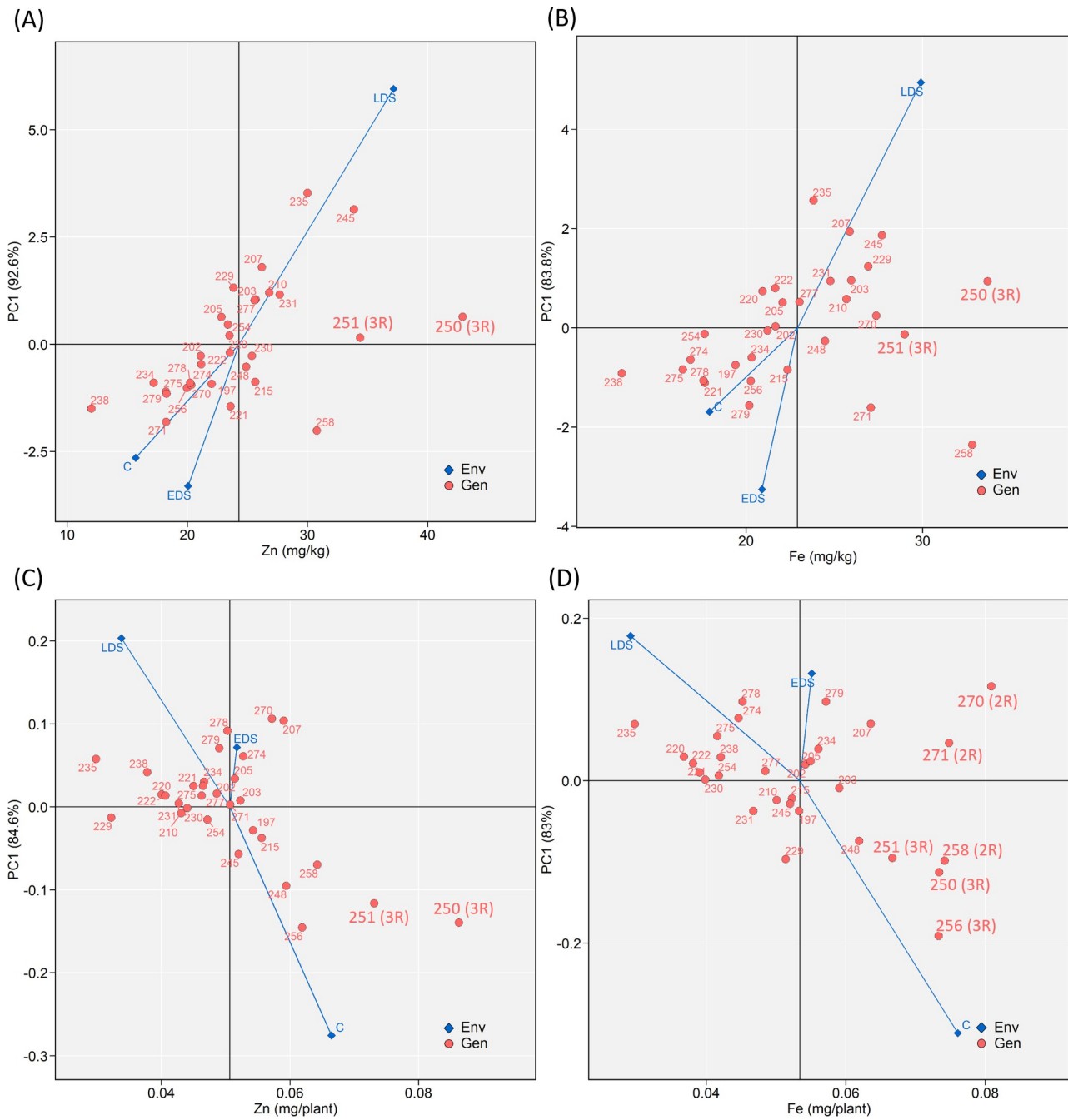

**Fig 6.** Additive main effects and multiplicative interaction (AMMI) biplots showing (A) Zn concentration, (B) Fe concentration, (C) Zn yield and (D) Fe yield versus the first principal component (PC1) score of 30 genotypes (Gen) and three growing conditions (Env) including control (abbreviated as C), early drought stress (EDS) and late drought stress (LDS). Genotypes located closer to the horizontal axis (score 0 on PC1) are those showing relatively higher stability across the three growing conditions. The vertical line in each figure indicates the average Zn and Fe grain concentration and mineral yield of the 30 genotypes.

## Stability of the concentration and yield of Se

The strongest genotype × environment interaction forces were identified under LDS and control for Se concentration and Se yield, respectively, and both resulted in above-average values (Fig 8). A more scattered distribution of the above-average genotypes was found for Se

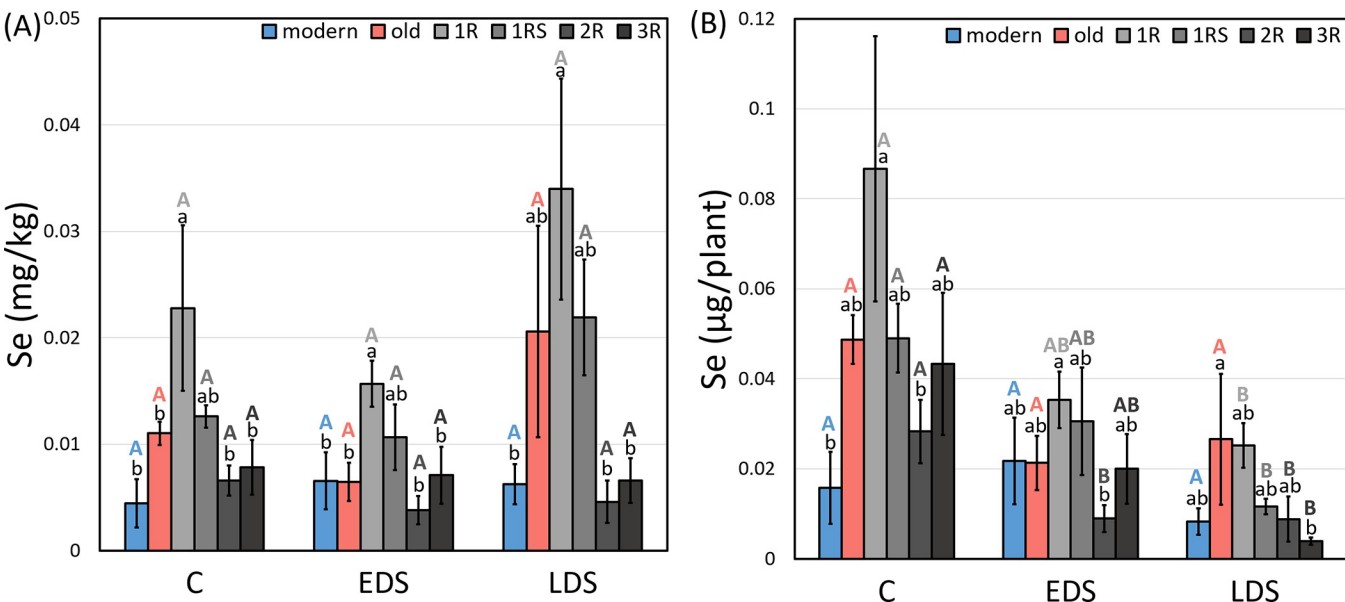

**Fig 7.** The mean (A) Se grain concentration and (B) Se mineral yield of each genotype group under control (abbreviated as C), early drought (EDS) and late drought stress (LDS). Modern = approved cultivars and breeding lines received from company Lantmännen, old = old Swedish cultivars released from 1928 to 1990, 1R, 1RS, 2R and 3R = introgressions of chromosome 1R, 1RS, 2R and 3R. Means of the same genotype group between treatments marked by the same capital letters do not differ significantly. Means between different genotype groups within each treatment marked by the same lower letters do not differ significantly (LSD post-hoc test at p < 0.05).

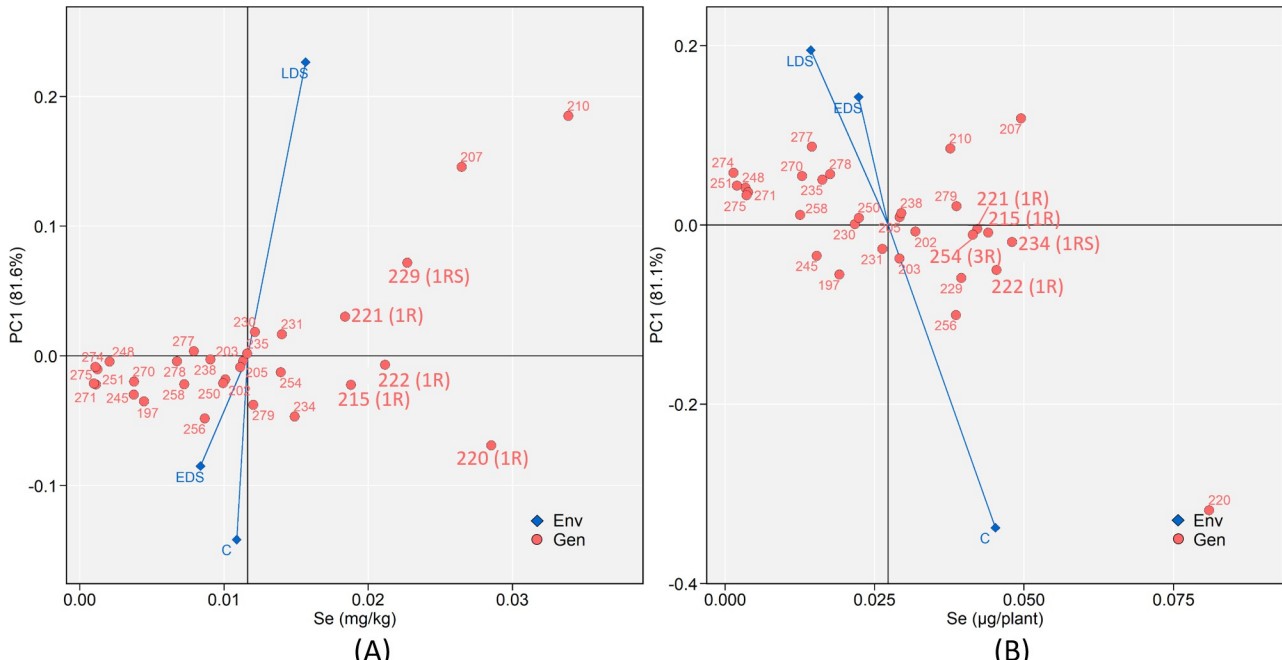

**Fig 8.** Additive main effects and multiplicative interaction (AMMI) biplots showing (A) Se grain concentration and (B) Se mineral yield versus the first principal component (PC1) score of 30 genotypes (Gen) and three growing conditions (Env) including control (abbreviated as C), early drought stress (EDS) and late drought stress (LDS). Genotypes located closer to the horizontal axis (score 0 on PC1) are those showing relatively higher stability across the three growing conditions. The vertical line in each figure indicates the average Se grain concentration and mineral yield of the 30 genotypes.

concentration compared to the below-average genotypes (Fig 8A), indicating a clear dispersive effect from LDS. The genotypes 215 (1R), 220 (1R), 221 (1R), 222 (1R) and 229 (1RS) were identified as the high-and-stable genotypes for Se concentration (Fig 8A) while 215 (1R), 221 (1R), 222 (1R), 234 (1RS) and 254 (3R) were identified as the high-and-stable genotypes for Se yield (Fig 8B).

## Concentration of Ca, Cu, K, Mn, Mg, Na, P and S

No effect of EDS or LDS was found on Ca concentration. 1RS genotypes showed a significantly higher Ca concentration than modern and old genotypes under control while 1RS genotypes maintained a higher Ca concentration than modern genotypes under LDS (Fig 9A).

An increase in Cu concentration was noted for 1R genotypes under EDS as compared to control, while LDS resulted in increases in Cu concentrations for modern, old and 1R genotypes. The 3R genotypes showed a higher Cu concentration than 1R, modern and old genotypes under control while no difference was found between genotypes groups under EDS and LDS (Fig 9B).

No effect from EDS was found on K concentration while significant increases were noted for modern, 1R, 1RS and 3R genotypes under LDS. Old genotypes showed a significantly higher K concentration than 2R genotypes under control while under EDS, old genotypes showed a higher value than 1R, 2R and 3R genotypes, and 1RS showed a higher value than 2R genotypes. No variation was found between genotype groups under LDS (Fig 9C).

Increases in Mn concentration were found for modern and 3R genotypes under EDS as compared to control while no effect of LDS was observed. The 1R genotypes showed high Mn concentration under control (higher than modern, 1RS, 2R and 3R) and EDS (higher than modern, old, 1RS and 3R) while under LDS, 1R genotypes only showed a significantly higher Mn concentration than 1RS and 3R genotypes (Fig 9D).

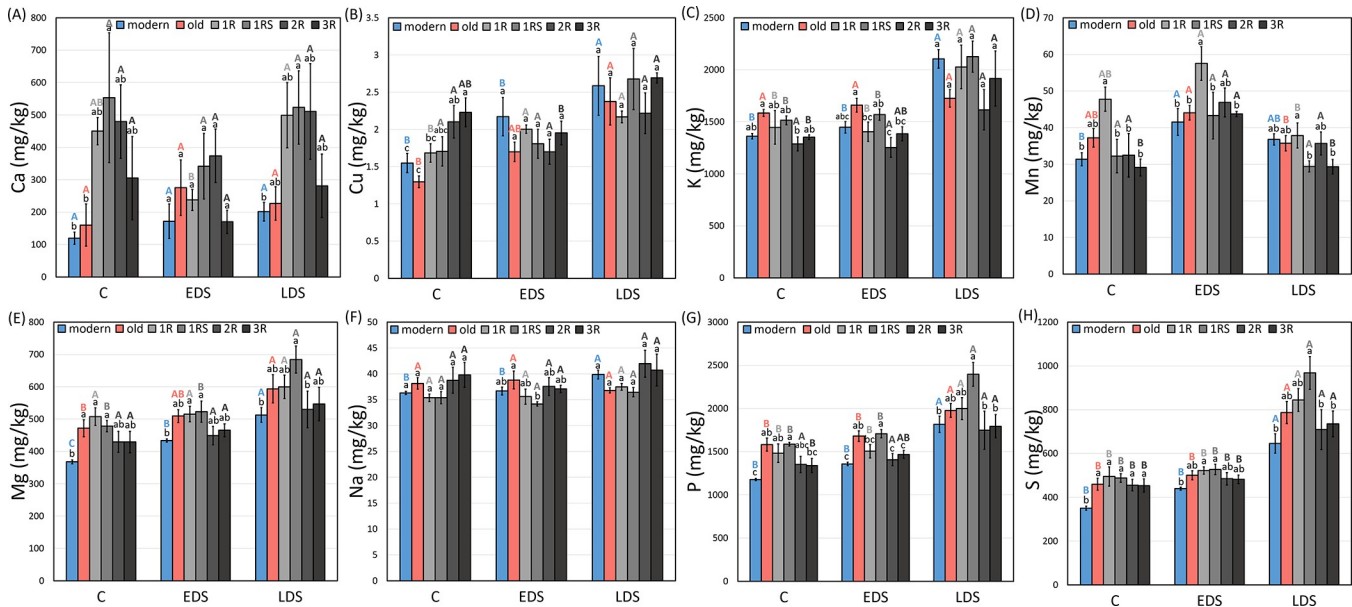

**Fig 9.** The mean (A) Ca, (B) Cu, (C) K, (D) Mn, (E) Mg, (F) Na, (G) P, (H) S concentration of each genotype group under control (C), early drought (EDS) and late drought stress (LDS). Modern = approved cultivars and breeding lines received from company Lantmännen, old = old Swedish cultivars released from 1928 to 1990, 1R, 1RS, 2R and 3R = introgressions of chromosome 1R, 1RS, 2R and 3R. Means of the same genotype group between treatments marked by the same capital letters do not differ significantly. Means between different genotype groups within each treatment marked by the same lower letters do not differ significantly (LSD post-hoc test at p < 0.05).

A high Mg concentration was found for modern genotypes under control while under LDS, increases were noted in modern, old and 1RS genotypes. Modern genotypes showed a generally low Mg concentration under control (lower than old, 1R and 1RS) and EDS (lower than 1R and 1RS) while 1RS genotypes showed high concentrations (higher than modern and 2R) under LDS (Fig 9E).

The drought effect on Na concentration was only found in modern genotypes under LDS. The only difference between genotype groups was noted under EDS where old genotypes showed a higher Na concentration than 1RS genotypes (Fig 9F).

No effect from EDS was found on P concentration while under LDS, significant increases were noted for modern, old, 1R, 1RS and 3R genotypes. Under control, modern genotypes showed a lower P concentration than old, 1R and 1RS genotypes, and 1RS genotypes showed a higher value than 3R genotypes. Under EDS, 1RS genotypes showed a higher P concentration than modern, 1R, 2R and 3R genotypes, and old genotypes showed a higher value than modern, 2R and 3R genotypes. 1RS genotypes maintained a higher P concentration than modern, 2R and 3R genotypes under LDS (Fig 9G).

No effect from EDS was found on S concentration while all the genotype groups showed an increase under LDS. Significantly the lowest S concentration was found for modern genotypes under control. 1R and 1RS genotypes showed a higher S concentration than modern genotypes under EDS while 1RS showed a higher value than modern, 2R and 3R genotypes under LDS (Fig 9H).

## Yield of Ca, Cu, K, Mn, Mg, Na, P and S

A decrease in Ca yield was found for 1R genotypes under EDS while 1R and 1RS genotypes showed a decrease under LDS. 1RS genotypes showed a significantly higher Ca yield than modern genotypes under control while 2R genotypes showed a higher value than modern, old, 1RS and 3R genotypes under LDS (Fig 10A).

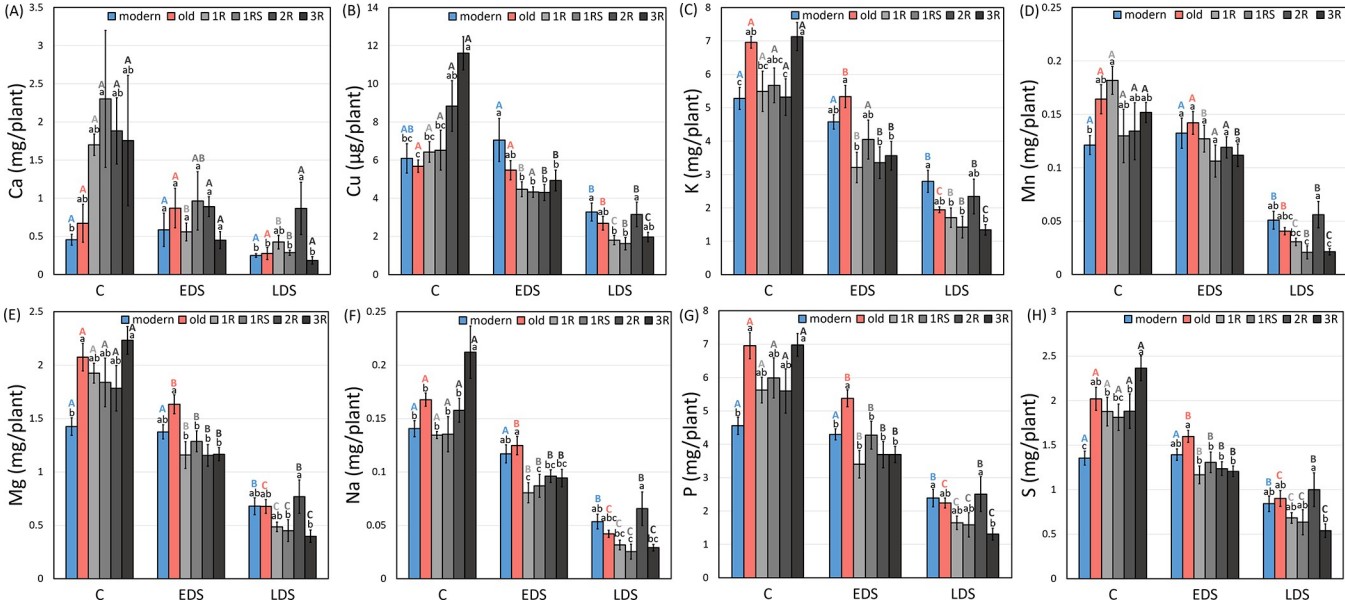

**Fig 10.** The mean (A) Ca, (B) Cu, (C) K, (D) Mn, (E) Mg, (F) Na, (G) P, (H) S mineral yield of each genotype group under control (C), early drought (EDS) and late drought stress (LDS). Modern = approved cultivars and breeding lines received from company Lantmännen, old = old Swedish cultivars released from 1928 to 1990, 1R, 1RS, 2R and 3R = introgressions of chromosome 1R, 1RS, 2R and 3R. Means of the same genotype group between treatments marked by the same capital letters do not differ significantly. Means between different genotype groups within each treatment marked by the same lower letters do not differ significantly (LSD post-hoc test at p < 0.05).

The Cu yield of 1R, 2R and 3R genotypes was significantly decreased by EDS while the Cu yield of most genotype groups was decreased by LDS except for modern genotypes. 3R genotypes showed a significantly higher Cu yield than modern, old, 1R and 1RS genotypes under control while modern genotypes showed a higher value than all the introgression genotypes under EDS. A higher Cu yield was noted for modern and 2R genotypes than 1R and 1RS genotypes under LDS (Fig 10B).

Old, 1R, 2R and 3R genotypes showed a decrease in K yield under EDS. LDS significantly decreased the K yield of all the genotype groups with a more severe impact on modern, old, 1RS and 3R genotypes. 3R genotypes showed a higher value than modern, 1R and 2R genotypes while old genotypes showed a higher value than modern and 2R genotypes under control. A higher K yield was found for old genotypes than 1R, 2R and 3R genotypes under EDS while a higher value was found for modern genotypes than 1R, 1RS and 3R genotypes under LDS (Fig 10C).

A decrease in Mn yield was noted for 1R and 3R genotypes under EDS while all the genotype groups showed a decrease (more profound on 1R and 3R than EDS) under LDS. 1R genotypes showed a higher Mn yield than modern genotypes under control. 2R genotypes showed a higher value than the rest of introgression genotype groups while modern genotypes showed a higher Mn yield than 1RS and 3R genotypes under LDS (Fig 10D).

Mg, Na, P and S shared the same pattern of drought effects. Only modern genotypes maintained Mg, Na, P and S yield under EDS while LDS significantly decreased the Mg, Na, P and S yield of all the genotype groups with a more severe impact on modern, old, 1R, 1RS and 3R genotypes than EDS (Fig 8E–8H). Modern genotypes showed a lower Mg yield than old and 3R genotypes under control. Old genotypes showed a higher Mg yield than all the introgression genotypes under EDS while 2R genotypes showed a higher Mg yield than 1RS and 3R genotypes (Fig 10E).

Significantly the highest Na yield was found for 3R genotypes under control. Old genotypes showed a higher value than all the introgression genotypes while modern genotypes showed a higher value than 1R and 1RS under EDS. 2R genotypes showed a higher Na yield than the other introgression genotype groups under LDS (Fig 10F).

Modern genotypes showed a lower P yield than old and 3R genotypes under control. Significantly the highest P yield was found for old genotypes under EDS while modern and 2R genotypes showed a higher P yield than 3R under LDS (Fig 10G).

Modern genotypes showed a lower S yield than old, 1R, 2R and 3R genotypes while 3R genotypes showed a higher S yield than the other introgression genotypes under control. A higher value was found for old genotypes than all the introgression genotypes under EDS while 2R genotypes showed a higher S yield than 3R genotypes (Fig 10H).

**Table 1. Genotypes with high and stable grain concentration and mineral yield of eight elements (Ca, Cu, K, Mn, Mg, Na, P and S) identified by additive main effects and multiplicative interaction (AMMI).**

|    | High and stable mineral concentration (S2 Fig) | High and stable mineral yield (S3 Fig) |
|----|-----------------------------------------------|----------------------------------------|
| Ca | 258 (2R), 270 (2R), 271 (2R) | 258 (2R), 270 (2R), 271 (2R) |
| Cu | 245 (2R), 250 (3R), 251 (3R) | 234 (1RS), 254 (3R), 256 (3R) |
| K | 202 (old), 210 (1R), 215(1R), 229 (1RS), 235 (1RS) | 202 (old), 203 (old), 205 (old), 207 (old), 238 (2R) |
| Mn | 220 (1R), 222 (1R), 258 (2R) | 202 (old), 203 (old), 220 (1R) |
| Mg | 203 (old), 210 (1R), 231 (1RS), 245 (2R) | 203 (old) |
| Na | 207 (old), 256 (3R), 258 (2R) | 207 (old) |
| P | 203 (old), 210 (1R), 215 (1R), 229 (1RS), 230 (1RS). | 202 (old), 203 (old), 205 (old), 234 (1RS) |
| S | 203 (old), 210 (1R), 215 (1R) | 203 (old), 234 (1RS), 238 (2R) |

## Stability of Ca, Cu, K, Mn, Mg, Na, P and S

Genotypes identified by AMMI with high and stable mineral concentration (S2 Fig) and mineral yield (S3 Fig) across three treatments are listed in Table 1. Basically, Ca and Cu were dominated by 2R and 3R genotypes respectively. The high and stable concentrations of K and Mn were dominated by 1R and/or 1RS genotypes while old genotypes dominated the yield of K and Mn. No clear pattern was found for Mg and Na concentration, and only one old genotype was identified for Mg and Na yield. The high and stable concentrations of P and S were dominated by 1R and/or 1RS genotypes. Old genotypes dominated high and stable P yield while no clear pattern was found for S yield. None of the modern genotypes was identified for high stability and performance (Table 1, S2 and S3 Figs).

## Discussion

This study clearly showed the contrasting performance of different sources of wheat germplasm in terms of mineral accumulation under drought stress. Genotypes with chromosome 3R demonstrated strong Zn and Fe uptake as well as high Cu, K, Na and S yield under control treatment. Several genotypes containing 3R were also identified with stable Zn and Fe accumulation from well-watered to two types of drought conditions. High and stable Se accumulation was found specifically in 1R genotypes while modern lines showed a lower accumulation in most of the studied minerals compared to other genotype groups. Old Swedish cultivars (K, Mg, Na, P and S) and introgression lines with 2R (Fe, Ca, Mn, Mg and Na) displayed outstanding tolerance in terms of mineral yield to EDS and LDS, respectively.

The significantly higher Zn and Fe accumulation found in 3R genotypes than most of the other genotype groups under control treatment suggested the presence of genes on chromosome 3R contributing to increases in the most human-health-related mineral nutrients Zn and Fe. An inadequate intake of Zn and Fe has been reported to cause a series of diseases related to liver function, diarrheal and immune system [3,4]. Interestingly, the increases in Zn and Fe yield were more significant than the increases in concentration, which might result from the high grain yield of 3R genotypes, as has been reported previously [34]. Thus, on top of the concentration increases in Zn and Fe, the total amount of Zn and Fe provided by a single wheat plant was further enhanced by positive effects on grain yield by the 3R. Furthermore, some 3R genotypes (250 and 251) were found with simultaneous high Zn and Fe levels across C, EDS and LDS conditions, proving the stability in Zn and Fe levels of these lines. The positive relationship between the grain content of Zn and Fe identified in this study, corresponds with earlier reports [38,39], thereby suggesting opportunities to breed for both these minerals in parallel. In addition to Zn and Fe, a considerably high yield of Cu, K, Na and S was also observed in 3R genotypes, which further consolidated the crucial role of 3R in increasing the nutritional value of the wheat grain and flour products. Previous studies on the functions of 3R in wheat have covered aspects such as the strengthened resistance to stem rust [40,41], grain protein content [42], tolerance to drought [34] and aluminum stress [43]. Attempts to improve grain nutrients content using chromosomes from wheat alien species have been made in a number of studies because of the two to three times higher Zn and Fe content observed in wild relatives as compared to modern wheat [44–46]. However, reports about the effect of rye chromosome 3R on the nutritional value of wheat are lacking. 3R has been reported with a significantly lower transmission rate (25.0%) than 1R (51.6%) and 2R (51.6%) during backcrossing [47], which might have hindered its wide use in wheat breeding. The potential of 3R demonstrated in this study suggests that it should be better exploited as a critical germplasm resource for the biofortification of wheat against the global problem of malnutrition (especially Zn and Fe deficiencies). In addition to the improved mineral concentration, wheat-rye

introgression lines carrying chromosome 3R have also been found with a significantly higher grain protein concentration, especially compared to 1R, 1RS and 2R genotypes [42]. Effective methods should be explored to increase the rate of successful rye-to-wheat 3R transfer while genes responsible for nutritional value should be explicitly searched on chromosome 3R.

Se is another health-related essential mineral nutrient for humans and its deficiency has extended to a population of one billion worldwide [48,49] while the effect on wheat is known to be strongly dependent on the environment [50]. In this study, contrasting Se accumulation between different genotype groups was observed. Across all three treatments, 1R genotypes showed a significantly higher Se concentration than other genotype groups, except old (LDS) and 1RS, suggesting that the positive effect of chromosome 1R on grain Se concentration might withstand drought stress. This was confirmed by AMMI analysis where 1R genotypes (215, 220, 221, 222) dominated with high and stable Se concentration and yield. Rye and especially chromosome 1R, has been used as a good genetic source in wheat breeding for different purposes. Genes (e.g. *Sr31*, *Yr9*, *Lr26* and *Pm8*) present on 1R have been largely exploited for disease resistance (stem rust, stripe rust, leaf rust and powdery mildew, respectively) in wheat [51,52]. Furthermore, 1R has also been reported to be responsible for improved root traits in wheat [53,54]. In a direct comparison between wheat and rye, a 35% higher grain Se concentration was obtained in field-grown rye as compared to synthetic hexaploid and tetraploid wheat, while a 40% higher foliar Se concentration was observed in hydroponic-grown rye as compared to two wheat landraces [55]. However, for plants, Se is an unessential mineral and its function is still not clear. The uptake mechanisms of inorganic Se are related to the two major chemical forms present in soil, i.e. selenate and selenite, as these are transferred by sulfate and phosphate transporters, respectively [56,57]. The high chemical similarity between Se and S might be the reason for them sharing the same set of transporters, which further explains the widely reported interaction that Selenate and sulfate compete in the process of plant uptake [58,59]. In addition to selenite and selenite, Se also exists in organic forms in soils e.g. seleno-glutathione and seleno-methionine [60]. Despite the fact that inorganic selenate is the most bioavailable form of soil Se, wheat plants have also been shown to actively take up Se from organic sources such as seleno-methionine [60]. Although wheat is the most efficient accumulator of Se compared to other common cereal crops [61], there are no available reports about the effect of 1R on the grain Se content in wheat. From a previous study, wheat lines introgressed with 1R displayed a robust early root vigor [34] which might benefit Se uptake from soil. Unlike chromosome 3R, 1R is not facing the difficulty of the poor transmission rate. Instead, it has already been widely used in wheat breeding, although a negative effect on baking quality is often coming along with the 1R introgression in wheat [62]. Our results suggested that in addition to traits like disease resistance and yield, research focuses on chromosome 1R should be shifted to its effect on wheat nutritional value, especially Se content.

Modern genotypes were found to have generally low concentrations in most studied minerals, which correspond with earlier studies [63] and this trend suggested its compromise in nutritional value during the pursuit of high yield. At the cost of yield-oriented breeding programs, modern wheat has suffered a decline in micronutrients content relative to the landraces, alien species and wild relatives, which has been described as the dilution effect resulting from the quick yield (starch content) increase [64–66]. In this study, modern genotypes showed significantly lower Mg, P and S concentrations than old genotypes and this old-to-modern downward trend has been noted in different studies [64,67]. However, the significantly different concentrations between old and modern genotypes were only observed in Mg, P and S, which might be because all the plants were grown in pots placed in an indoor controlled-environment chamber where the more robust root system of old genotypes did not get to play a role in accessing more nutrients. A larger variation in mineral concentrations

between modern and old wheat lines is expected in field conditions as has also been reported in previous studies [68,69]. In contrast, wheat-rye introgression genotypes showed a superior performance as significantly higher concentrations of Zn, Fe, Se, Ca, Cu, Mn, Mg, P and S were found in at least one of the introgressed genotype groups as compared to modern genotypes. This finding agreed with a previous study where increased levels of minerals (especially Zn and Fe) were obtained in introgressed genotypes with 1R, 2R or 5R [70]. Interestingly, the modern genotypes evaluated here also showed a lower grain protein concentration than introgressed genotypes in our previous study using a larger set (a total of 73) of genotypes [42]. Thus, rye chromosomes can be used as a strong alien genetic source to elevate both the mineral and protein contents that have been compromised in modern wheat, and therefore to fulfill the rising awareness of nutritional quality in foods.

The drastic yield decrease of wheat plants under drought stress usually consequently leads to an increase in mineral concentrations [71]. Differently, mineral yield gives the amount of minerals provided by a single plant, and therefore, it is a more suitable parameter to evaluate plants' drought tolerance in terms of minerals (mineral-yield maintaining ability). In our results, old genotypes showed outstanding tolerance to early drought due to their relatively high yield in K, Mg, Na, P and S, while 2R genotypes were found well performed in Fe, Ca, Mn, Mg and Na under late drought stress. Old wheat has been reported to show a high accumulation of minerals in several studies [64,68,72]. The genome of old Swedish cultivars might contain some ancestral genes that have been lost during decades of human selection, and reclaiming those genes gives a better opportunity in selections for tolerant lines. To our best knowledge, there are only two studies that mentioned the effect of chromosome 2R on increasing mineral concentrations in wheat [70,73]. The genomes of alien relatives to wheat are known for their broad spectrum of biotic/abiotic resistance [41,70]. Thus, with respect to mitigating the impact of drought on the nutritional value of wheat, genes responsible for tolerance to early drought and late drought should be searched in genomes of old Swedish cultivars and introgressions with chromosome 2R, respectively.

## Conclusion

In the context of global climate change, food security is threatened by increasing drought events. More than two billion people across the globe are suffering from micronutrient deficiencies caused by the consumption of a nutrient-poor diet. Therefore, improvements in the nutritional value of wheat, one of the three major crops, are urgently needed. The nutritional value of wheat is largely determined by its mineral composition, especially the Zn, Fe and Se content. Here, chromosome 3R, introgressed to wheat, contributed to a high mineral yield of Zn, Fe, Cu, K, Na and S under controlled cultivation conditions, thereby demonstrating the strong role of 3R for an increase of the total amount of nutrients in wheat grown under favorable conditions. Furthermore, the 3R genotypes 250 and 251, contributed to a high and stable concentration and yield of both Zn and Fe under drought conditions, suggesting these lines as effective genetic resources to be used in breeding for high contents of both these nutrients in parallel and also for stability across climate change conditions. A high and stable performance for Se of 1R genotypes indicated a potential for the use of chromosome 1R in breeding to increase the Se efficiency of wheat. Old Swedish cultivars and introgressed 2R genotypes demonstrated tolerance to early drought and late drought, respectively, by a high mineral yield, which also resulted in significant stability across drought treatments for these lines. Thus, 3R and 1R genotypes are proposed as the two potential gene pools related to Zn/Fe and Se content across climate change environments, while old Swedish cultivars and 2R genotypes were identified as germplasms for stable mineral supplies under drought conditions. Breeding strategies

should be adjusted accordingly to biofortify wheat nutritional values, as these values have been reduced in modern wheat.

## Supporting information

**S1 Fig.** Zn (A-C) and Fe (D-E) concentration of each genotype under control (C), early drought stress (EDS) and late drought stress (LDS). Modern = approved cultivars and breeding lines received from company Lantmännen; old = old Swedish cultivars released from 1928 to 1990; 1R, 1RS, 2R and 3R = Introgressions of chromosome 1R, 1RS, 2R and 3R. The value of each genotype was generated from the mean of three biological replicates.
(TIF)

**S2 Fig.** Additive main effects and multiplicative interaction (AMMI) biplots showing concentration of (A) Ca, (B) Cu, (C) K, (D) Mn, (E) Mg, (F) Na, (G) P and (H) S versus the first principal component (PC1) score of 30 genotypes (Gen) and three growing conditions (Env) including control (abbreviated as C), early drought stress (EDS) and late drought stress (LDS). Genotypes located closer to the horizontal axis (score 0 on PC1) are those showing relatively higher stability across the three growing conditions. The vertical line in each figure indicates the average mineral concentration of the 30 genotypes.
(TIF)

**S3 Fig.** Additive main effects and multiplicative interaction (AMMI) biplots showing mineral yield of (A) Ca, (B) Cu, (C) K, (D) Mn, (E) Mg, (F) Na, (G) P and (H) S versus the first principal component (PC1) score of 30 genotypes (Gen) and three growing conditions (Env) including control (abbreviated as C), early drought stress (EDS) and late drought stress (LDS). Genotypes located closer to the horizontal axis (score 0 on PC1) are those showing relatively higher stability across the three growing conditions. The vertical line in each figure indicates the average mineral yield of the 30 genotypes.
(TIF)

**S1 Table. Information about genotypes used in the present study.**
(XLSX)

**S2 Table. Data of mineral concentration, grain yield and mineral yield of each genotype.**
(XLSX)

**S3 Table. ANOVA table in the form of mean square values for minerals under different drought stress conditions (\*\*\*: sig. < 0.001, \*\*: sig. < 0.01, \*: sig. < 0.05).**
(XLSX)

## Acknowledgments

Special thanks to Maria Luisa Prieto-Linde for providing significant technical assistance throughout this study.

## Author Contributions

**Conceptualization:** Yuzhou Lan, Ramune Kuktaite, Aakash Chawade, Eva Johansson.

**Data curation:** Yuzhou Lan.

**Formal analysis:** Yuzhou Lan.

**Funding acquisition:** Eva Johansson.

**Investigation:** Yuzhou Lan.

**Methodology:** Yuzhou Lan, Ramune Kuktaite, Aakash Chawade, Eva Johansson.

**Project administration:** Eva Johansson.

**Resources:** Eva Johansson.

**Supervision:** Ramune Kuktaite, Aakash Chawade, Eva Johansson.

**Validation:** Yuzhou Lan.

**Visualization:** Yuzhou Lan.

**Writing – original draft:** Yuzhou Lan.

**Writing – review & editing:** Yuzhou Lan, Ramune Kuktaite, Aakash Chawade, Eva Johansson.

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
