## [Decision Letter · Decision Letter 0]

11 Dec 2023

PONE-D-23-34764High and stable mineral composition in diverse spring wheat lines – sources for biofortification in a changing climatePLOS ONE

Dear Dr. Johansson,

Thank you for submitting your manuscript to PLOS ONE. After careful consideration, we feel that it has merit but does not fully meet PLOS ONE’s publication criteria as it currently stands. Therefore, we invite you to submit a revised version of the manuscript that addresses the points raised during the review process.

We look forward to receiving your revised manuscript.

Kind regards,

Sindhu Sareen

Academic Editor

PLOS ONE

Journal Requirements:

"Trees and Crops for the Future (TC4F) 

SLU Grogrund"

Reviewers' comments:

Reviewer's Responses to Questions

**Comments to the Author**

1. Is the manuscript technically sound, and do the data support the conclusions?

Reviewer #1: No

Reviewer #2: Yes

2. Has the statistical analysis been performed appropriately and rigorously? 

Reviewer #1: Yes

Reviewer #2: Yes

3. Have the authors made all data underlying the findings in their manuscript fully available?

Reviewer #1: No

Reviewer #2: Yes

4. Is the manuscript presented in an intelligible fashion and written in standard English?

Reviewer #1: Yes

Reviewer #2: Yes

5. Review Comments to the Author

Reviewer #1: The work is extremely interesting and valuable but to my opinion there are some drawbacks.

Major comments

Material and Methods section is extremely short and should be seriously revised. The authors suggest the readers to read their previous article published in another journal!!!!~ Such a proposal may be OK only in case the previous work has been published in the same journal. Please, add full description of the experiment, soil characteristics, Se, Fe and Zn content in soil, the dates of seeds sowing and harvest, the number of genotypes in each group studied, drought treatment, etc.

1) Use Italics for the Latin name of wheat

2) The manuscript is overloaded with abbreviations which hinders the perception of the data. Please, don’t abbreviate at least ‘control’ in the text and change ‘C’ to ‘control’. Abbreviations are useful in Figures but extremely tedious in the text.

3) Line 102 mineral determination: what was the internal and external reference standards for the elements tested?

4) Line 42 I can’t agree with the statement that ‘sulfur (S) is mainly supplied to humans in the form of methionine’. What about cysteine, sulfur derivatives in Allium species, glucosinolates in Brassica plants???

5) The title does not provide a concrete information about the content of the work. May be, it is better to change it to ‘The impacts of early and late drought stress on wheat grain mineral composition”?

6) It is highly desirable to add grain yield for different genotypes and genotype groups

7) Is there any information about the differences in grain quality except miberals?

8) While speaking about the relationship between the elements (line 203, Zn, Fe) the significance of the correlation should be indicated. Please, add ‘p<’

9) line 425 ‘’ The uptake mechanisms of Se are related to the two major chemical forms present in soil, i.e. selenate and selenite, as these are transferred by sulfate and phosphate transporters, respectively- what about organic Se and Se-S interaction?

10) The authors have forgotten about the work of Ayed, S.; Bouhaouel, I.; Othmani, A.: Screening of Durum Wheat Cultivars for Selenium Response under Contrasting Environments, Based on Grain Yield and Quality Attributes. Plants 2022, 11, 1437. https://doi.org/10.3390/ plants11111437. Please add this citation in Introduction/Discussion sections and in the reference list

Minor comments:

11) Line 490: a misprint: ‘aluminium’ change to ‘aluminum’

Reviewer #2: The manuscript is technically sound, and the presented data support the conclusions. Appropriate and rigorous statistical analysis is performed. Moreover, the manuscript is presented in an intelligible fashion and written in standard English. Thus, I confidently recommend to accept the manuscript for publication in POLS ONE.

6. PLOS authors have the option to publish the peer review history of their article (what does this mean?). If published, this will include your full peer review and any attached files.

Reviewer #1: No

Reviewer #2: **Yes: **K.M. Mohiuddin

---

## [Author Response · Author response to Decision Letter 0]

15 Jan 2024

Dear Editor,

We have now gone through the manuscript in relation to specific editor and reviewer comments;

1. We have made sure that the manuscript meets PLOS ONE´s style requirements, including them for file naming.

3. Data from this study have now been included as supplementary Table S2. There are no ethical or legal restrictions on sharing a de-identified data set.

4. ORCID iD of corresponding author have been included.

5. The reference list has been reviewed to ensure that it is correct.

Answers to the reviewers comments are found in the point-by-point letter.

---

## [Editor Report · Decision Letter 1]

23 Jan 2024

Chasing high and stable wheat grain mineral content: mining diverse spring genotypes under induced drought stress

PONE-D-23-34764R1

Dear Dr. Johansson,

We’re pleased to inform you that your manuscript has been judged scientifically suitable for publication and will be formally accepted for publication once it meets all outstanding technical requirements.

Kind regards,

Sindhu Sareen

Academic Editor

PLOS ONE
---

## [Editor Report · Acceptance letter]

7 Feb 2024

PONE-D-23-34764R1 

PLOS ONE

Dear Dr. Johansson, 

I'm pleased to inform you that your manuscript has been deemed suitable for publication in PLOS ONE. Congratulations! Your manuscript is now being handed over to our production team.

Kind regards, 

on behalf of

Dr. Sindhu Sareen 

Academic Editor

PLOS ONE